# Pre-Analytical Modification of Serum miRNAs: Diagnostic Reliability of Serum miRNAs in Hemolytic Diseases

**DOI:** 10.3390/jcm10215045

**Published:** 2021-10-28

**Authors:** Yukichi Takada, Tatsuki Shibuta, Mayu Hatano, Kenichi Sato, Mari Koga, Ayaka Ishibashi, Tetsuhiro Harada, Takashi Hisatomi, Hanae Shimura, Noriyasu Fukushima, Kamonlak Leecharoenkiat, Supat Chamnanchanunt, Saovaros Svasti, Suthat Fucharoen, Tsukuru Umemura

**Affiliations:** 1Department of Medical Technology and Sciences, International University of Health and Welfare, Okawa 831-8501, Japan; 1367053@g.iuhw.ac.jp (Y.T.); tshibuta@iuhw.ac.jp (T.S.); 1467061@g.iuhw.ac.jp (M.H.); ken-ichi_sato@iuhw.ac.jp (K.S.); shimura-afg@kuhs.ac.jp (H.S.); 2Clinical Laboratory, Kouhoukai Takagi Hospital, Okawa 831-8501, Japan; ma-koga@kouhoukai.org (M.K.); 16s1011@g.iuhw.ac.jp (A.I.); harada@kouhoukai.org (T.H.); 3Hematology, Kouhoukai Takagi Hospital, Okawa 831-8501, Japan; hisatomi@kouhoukai.org; 4Research Institute for Radiation Biology and Medicine, Hiroshima University, Hiroshima 734-8553, Japan; fukushin@hiroshima-u.ac.jp; 5Department of Clinical Microscope, Faculty of Medical Sciences, Chulalongkorn University, Bangkok 10330, Thailand; RBC_2524@hotmail.com; 6Faculty of Tropical Medicine, Mahidol University, Bangkok 100400, Thailand; supat.cha@mahidol.edu; 7Thalassemia Research Center, Institute of Molecular Biosciences, Mahidol University, Salaya, Nakhon Pathom 73130, Thailand; stssv@yahoo.com (S.S.); suthat.fuc@gmail.com (S.F.)

**Keywords:** microRNA, miR-451a, pre-analytical, blood drawing, serum separation, hemolytic disease

## Abstract

Circulating microRNAs (miRNAs) are useful biomarkers of hemolysis. Since blood cells are the main origins of circulating miRNAs, we evaluated blood cell-related pre-analytical modification of the miRNA signatures during blood drawing and serum processing. The levels of miRNA before and after ex vivo blood drawing were analyzed with the reverse transcriptase-based polymerase chain reaction method. Furthermore, the changes of miRNA signatures caused by different time-lag between blood drawing and serum preparation by 24 h were evaluated. Finally, we compared the miRNA levels between leftover samples and samples of hemolytic diseases. Blood drawing procedure induced increments of red blood cell (RBC)-related miRNAs (miR-451a, miR-486) about 2-fold. One hour standing of blood samples before serum separation induced almost the same increases in RBC-related miRNAs. To test the clinical usefulness of miR-451a as a biomarker of hemolytic diseases, we analyzed miRNAs of samples from 10 normal subjects, 30 leftover samples in the clinical laboratory, and 20 samples from patients with hemolytic diseases. Serum miR-451a significantly increased in patients with hemolytic anemia more than the levels of pre-analytical modification. In conclusion, the pre-analytical modification of serum miRNAs did not disturb the usefulness of RBC-derived miRNAs as biomarkers of hemolytic diseases.

## 1. Introduction

Circulating microRNAs (miRNAs) are expected as non-invasive relevant biomarkers of various diseases [1,2,3,4]. However, to develop miRNAs as practical biomarkers, specificity, predictability, robustness, translatability is recommended [4]. Hemolytic diseases are caused by accelerated RBCs destruction classified as intravascular, extravascular hemolysis, and ineffective erythropoiesis [5]. When hemolysis occurs, many components are released into the blood or bone marrow space. Erythroid cell-specific materials such as hemoglobin (Hb) or nucleic acids are important biomarkers of normal or dysregulated erythropoiesis. Many genes and miRNAs regulate the proliferation and differentiation of erythroid cells [6,7]. Expression of miR-451a is limited only to the erythroid lineage and is critical to initiate late erythropoiesis [5,8,9]. Thus, Hb and miR-451a are good biomarkers to detect hemolytic diseases [10,11,12,13,14,15]. 

Hemolytic diseases are classified as intravascular hemolysis, extravascular hemolysis, or ineffective erythropoiesis caused by premature decay of erythroblasts in the bone marrow. The main population of erythroid cells in the bone marrow is erythroblasts which are nucleated and proliferate to produce RBCs [5,6,7]. Although circulating RBCs are enucleated cells with no cellular division, they still contain a small amount of ribosomal RNAs, short RNAs, and miRNAs [16]. miR-451a is one of such nucleic acids, of which expression is highly restricted to the late erythropoiesis [17,18,19]. Knockout of miR-451a in zebrafish or mouse models resulted in the loss of RBC production [20,21,22]. Circulating RBCs contain abundant miR-451a, miR-486 and miR-16 [14,15,23,24]. Previous studies have shown elevated levels of circulating mR-451a in hemolytic diseases [12,13]. On the other hand, several groups pointed out the possibility that pre-analytical hemolysis may disturb the utility of serum miR-451 as a hemolysis biomarker [10,25,26].

The process of blood sampling has several factors to modify blood components, for example, time of sampling, duration of venous clipping, mechanical damage of blood vessel or blood components, a time-lag before serum or plasma separation, preservation [11,27,28,29,30]. For producing stable miRNA results as a clinical application, standardization of pre-analytical steps is highly recommended. About 70% of errors in clinical medicine are thought to be preanalytical. Destruction of RBCs in samples, namely ex vivo hemolysis, is the main preanalytical error [12]. Almost 90% of circulating RNAs is miRNA [31]. Blood cells are major sources of circulating miRNAs [23]. Therefore, the release of cellular miRNAs in blood cells to serum fraction during or after blood drawing may modify miRNA signature in samples. 

There are few studies comparing the levels of circulating miRNAs between pre-analytical modification and pathophysiological state [10,11,12,13,14,15]. In this study, we focused on the pre-analytical modification of serum erythroid-specific miRNAs as biomarkers for hemolytic diseases.

## 2. Materials and Methods

In total, 145 samples were examined: 116 leftover sera in the clinical laboratory of Kouhoukai Takagi Hospital, 8 control subjects, 2 autoimmune hemolytic anemia, 13 beta-thalassemia, 1 alpha-thalassemia, 3 malaria (Plasmodium falciparum), were studied. Serum samples were collected using the standard methods according to the guidelines by the Japanese Association of Medical Technologists (JCCLS) and joint EFLM-COLABIOCLI Recommendation [32,33]. Briefly, whole blood was drawn using a vacuum sample tube containing coagulation stimulators. Serum was prepared after 30 min at room temperature, using centrifugation of 3500 rpm, 10 min. After passing a membrane filter with 0.45 μm pore size (Sartrius, Göttingen, Germany), serum was kept at −80 °C for 7 days until analysis in the case of leftover samples. Sera from control subjects and patients with hemolytic anemia were processed immediately after blood drawing and were preserved using the same methods. For detection of artifact hemolysis during blood drawing, the first blood samples were collected into the tubes containing EDTA. Half of the first blood sample was drawn from the initial vacuum tube into the second vacuum tubes containing EDTA. Then, plasma was obtained from each tubes using the same procedures as serum preparation. miRNAs in the paired aliquots were analyzed in the same assay. The hemolysates with sequentially diluted concentrations of hemoglobin were obtained by hemolysis of packed red blood cells (PRBCs). Briefly, whole blood was collected into an EDTA-containing tube, and PRBCs were made with centrifugation of 3500 rpm for 10 min. Then, hemolysate was obtained using freeze-thawing after preservation at −80 °C for overnight. Concentrations of Hb in each diluted lysates were measured with two different methods, the first: the method spectrophotometric method (U-5100, HITACHI Co., Tokyo, Japan) using 540 nm OD to detect free Hb, and the second: NanoDrop equipment (ND 2000c, ThermoFisher Scientific Co. Waltham, MA, USA,) using 414 nm OD to detect oxyhemoglobin [34]. Finally, we used the oxyhemoglobin method due to its higher sensitivity (Figure 1a).

Serum miRNAs were analyzed using the reverse transcriptase-based quantitative polymerase chain reaction (RT-qPCR) method [13,35]. MiRNAs were extracted using the Nucleospin miRNA Plasma Extraction Kit (Takara, Kusatsu, Japan) according to the manufacturer’s instructions. During the extraction process, one femtomole of cel-miR-39 (CosmoBio, Tokyo, Japan) was added as a spiked-in control. Purified miRNAs were transcribed to complementary DNAs (cDNAs) using the High-Capacity cDNA Archive Kit (ThermoFisher Scientific Co., Waltham, MA, USA). cDNAs were amplified using the TaqMan miRNA Assay Kit (ThermoFisher Scientific Co., Waltham, MA, USA) and Master Mix (ThermoFisher Scientific Co., Waltham, MA, USA) [13,35]. The cycle condition of qPCR was instructed as enzyme activation at 95 °C for 10 min, PCR reaction at 95 °C for 15sec and 60 °C for 60 sec, repeated 40 cycles using real-time PCR equipment (ABI7500fast, ThermoFisher Scientific Co., Waltham, MA, USA). The expression levels of miRNAs were evaluated using the comparative threshold cycle (Ct) method [13,35]. Namely, the levels of each miRNA were compared after calculating ΔCt values: (Ct value of target miRNA)—(Ct value of cel-miR-39). This study has been complied with the institutional policies of The International University of Health and Welfare, Japan (Project No.:15—Ifh—102) and Mahidol University, Thailand (TMEC 20-055), in accordance the tenets of the Helsinki Declaration, and has been approved by the two universities ethical committee.

## 3. Results

### 3.1. miR-451a as a Biomarker of Hemolysis

By using a sequential dilution of hemolysates, we measured the concentration of miR-451a with RT-qPCR and free Hb with the oxyhemoglobin method (NanoDrop). The two parameters showed statistically significant correlation (r = 0.885, *n* = 3). Data indicated that miR-451a was sensitive enough to detect hemolysis (Figure 1).

### 3.2. The Effect of Blood Drawing on the miRNA Signature

We evaluated the influence of blood drawing on miRNA levels by using ex vivo drawing. Namely, each half volume of collected blood samples was drawn and transferred into the second vacuum tubes. To evaluate damages of each blood cell lineages, we selected six miRNAs: miR-451a, miR-486-5p, miR-16 for RBCs, miR-223-3p for granulocytes, miR-126 for platelets. Hb concentration increased 1.26 + 0.1-fold (*p* < 0.04). MiRNAs that are ubiquitous in RBCs increased more than other blood cell-derived miRNAs, miR-451a, 1.86 + 0.27 (mean + SEM, *n* = 8, *p* < 0.02), miR-486-5p, 1.85 + 0.28 (*p* < 0.02), miR-16, 1.97 + 0.32 -folds (*p* < 0.02) (Figure 2). miR-223, which is abundant in granulocytes, increased 1.39 + 0.31, platelet-related miR-126, 1.26 + 0.18 -folds, either were not statistically significant (Figure 2).

### 3.3. The Effect of Time-Lag before Serum Separation on the miRNA Signatures

We observed the change of five serum miRNAs (miR-451a, miR-486, miR-16, miR-223, and miR-126) at the different lag times after blood drawing at room temperature (Figure 3). miR-451a and miR-486 showed the earliest change, 1.62 and 1.65-fold by 1 h, 1.89 and 2.31-fold by 24 h, respectively. miR-16, which is contained in RBCs as well as in other cell lineages, increased 1.54-fold. miR-223, which is mainly contained in granulocytes, increased 1.81-fold at 24 h after sampling. miR-126, which is abundant in platelets, had a significant increment after 6 h (1.91-fold). Finally, all five miRNAs increased almost 2-fold by 24 h. The data indicate that lag-time-dependent modification of serum miRNA levels may be variable among the different blood cell types and one of the pre-analytical factors and that RBCs-derived miRNAs, miR-451a and miR-486, were most sensitive for lag-time before serum separation as same as the blood drawing step. Strict standardization is required for clinical application as the biomarker of hemolytic diseases.

### 3.4. Biochemical Indexes in Leftover Samples in Clinical Laboratory

The average level of oxyhemoglobin of leftover serum was 4.37 ± 0.19 (SEM, *n* = 116) (range: 0.77 to 12.3) mg/dL, and all 116 samples showed detectable hemolysis judged by 414 nm OD (NanoDrop method): the sensitivity of 0.59 mg/dL (Figure 4).

We measured the levels of LD, AST, and potassium which are used as biochemical markers for hemolysis, in each 116 blood samples. The average levels of three markers were LD, 162.8 ± 3.80 U/L (SEM, *n*=116), AST, 22.9 ± 1.52 U/L, and potassium, 4.2 ± 0.05 mmol/L (Table 1.). If we used the standard ranges of three markers, 120 to 220 U/L for LD, 10 to 35 U/L for AST, 3.5 to 4.5 mmol/L for potassium, 18 (16%), 16 (10%), and 29 (25%) leftover samples showed abnormally high levels, respectively. One sample showed extremely high levels of free Hb.

### 3.5. miR-451a Is Elevated in All Leftover Samples in Clinical Laboratory

We measured the miR-451a levels in 30 out of 116 leftover samples in clinical laboratory without obvious hemolysis. The average level of miR-451a in 30 samples was 16.2 ± 2.66 fmol/µL (SEM, *n* = 30), which was 2.6-fold higher with statistical significance (*p* < 0.05) than the average level of control subjects 6.28 ± 1.48 fmol/µL (SEM, *n* = 10) (Figure 5). Data show the significant increase in serum miR-451a far over pre-analytical modification of miR-451a levels in hemolytic diseases.

### 3.6. High Levels of Serum miR-451a in Hemolytic Diseases

Serum miR-451a levels were analyzed in 20 patients with hemolytic disease: 2 autoimmune hemolytic anemia, 3 malaria infections (*Plasmodium falciparum*), one alpha-thalassemia, 12 β-thalassemia, 2 paroxysmal nocturnal hemoglobinuria. All patients’ serum had higher miR-451a levels than the average ± 1 STDV (standard deviation) value of 30 leftover samples. Oppositely, only one leftover sample showed a higher level of miR-451a than the level of average—1 STDV (36.8 fmol/µL) of hemolytic anemia samples (Figure 5). This leftover serum had elevated values of LD (311 U/L), AST (59 U/L), free Hb (12.3 mg/dL).

## 4. Discussion

Previous studies showed variable results of miRNA might be induced by non-standardized pre-analytical processing of blood samples [25,36,37]. In this study, we focused on the pre-analytical modification of circulating miRNA signature for the purpose of standardization of circulating miRNA analysis.

Lippi et al. showed 65% of laboratory errors was caused by pre-analytical factor: hemolysis, inappropriate sample volume, the wrong container, clotting, contamination by infusion fluids or blood tube additives, storage, and freeze-thawing processes [12]. Dr. Kirschner’s group showed that hemolysis influenced the serum miRNA level, especially on miR-451a [10]. In this study, we focused on the pre-analytical steps after blood drawing, namely, the effect of mechanical damage of blood drawing on blood cells and the release of miRNAs from blood cells before serum preparation. Duration or pressure of tourniquet banding are also important factors to modify data. However, since we used leftover samples in the clinical laboratory except for control subjects in this study, the effect of factors before blood drawing was not examined. Blood drawing was conducted according to the standard procedure recommended in the guideline by the Japanese Committee for Clinical Laboratory Standards [33], which is similar to the European Federation of Clinical Chemistry and Laboratory Medicine (EFLM) recommendation [32]. Samples were collected in the same hospital under the same guideline, so that relatively equal conditions of the tourniquet were maintained.

Firstly, we evaluated the effect of blood drawing on levels of serum miRNAs with the ex vivo drawing from blood samples already taken (Figure 2). Our data indicated that blood drawing itself modified miRNA signature, mostly of RBC-derived miRNAs (miR-451a, miR-486-5p, miR-16): up to almost 2-fold increments (Figure 2). On the other hand, the degree of granulocytes and platelets damage by blood drawing itself was not significant enough to modify serum miRNA signature. The release of biochemical components such as LD or iron is a well-known phenomenon during blood drawing [5]. Our data have clearly shown that RBC-related miRNAs (miR-451a, miR-486) were also released to serum samples up to the two-fold level, indicating that blood drawing mainly modifies the miRNA signature of RBCs origin. RBCs are the most abundant cell population of blood cells (almost 40 to 45% of blood volume). Hemoglobin is an RBC-specific protein and consists of almost one-third of RBC weight. Thus, the detection of free hemoglobin has been a good biomarker of hemolysis. Since previous studies have shown that artificial hemolysis at the pre-analytical step is an unavoidable problem for accessing in vivo hemolysis [10,25,26,37], we examined the level of free Hb in 116 leftover serum samples taken for a routine laboratory test. Our study detected 4.37 ± 0.19 mg/dL of free oxyhemoglobin in 116 samples (Figure 4), which were almost the same level as previously reported (< 5 mg/dL) (Table 1) [11]. The free Hb level of more than 50 mg/dL is a clinical threshold of the presence of hemolysis, either by diseases or pre-analytical factors [11]. The miRNA signature was more sensitive to hemolysis even less than the upper limit of oxyhemoglobin detection (Figure 1) because of abundant miRNAs in RBCs [10,25,26,37]. Our data showed that clinical samples had 2.6-fold higher miR-451a levels than freshly processed control serum samples (Figure 5). This suggested that other factors besides blood drawing might produce different results.

Secondly, we tested the effects of time-lag between blood drawing and serum separation. Data indicated that RBC-related miRNAs (miR-451a and miR-486) are most sensitive to time lag and that levels in serum increased almost 2-fold from the initial level (Figure 3). Together with the effect of blood drawing, RBC-related miRNAs may increase 4-fold during pre-analytical steps. The other lineages of blood cells are reported to have a unique miRNA signature and contribute to the regulation of homeostasis and pathophysiology [38]. Granulocyte-derived miR-223 is reported as a miRNA regulating bacterial inflammation [39], acute lung injury [40], or wound healing [41]. Circulating miR-223 seems to be a good biomarker of these pathological states. miR-126-3p is one miRNA that is released from platelets and a possible biomarker of coagulation or thrombosis [42]. An increment of miR-223 was observed 24 h after sampling, platelet-related miR-126 after 3 h (Figure 3).

Our data suggest that for utilization of serum miRNAs as disease biomarkers, serum processing within one hour is recommended to avoid pre-analytical modification. Our data are similar to previously reported findings by Wu et al. who reported that miRNA signature was unchanged until 2 h after blood drawing [28]. The measurement of blood cell-related miRNAs may offer the possible quality control of samples and may contribute to reducing pre-analytical errors of laboratory tests [43].

Our data showed that the degree of pre-analytical modification was small to disturb diagnosis of hemolytic diseases which had higher miR-451a levels. In the case of hemolytic anemia, circulating mature RBCs are destructed by autoimmune mechanisms (autoimmune hemolytic anemia, paroxysmal nocturnal hemoglobinuria), membrane abnormalities, or α-thalassemia. miR-451a, miR-144, miR-486, and miR-16 are abundant in mature RBCs, so that accelerated destruction results in increased levels of those miRNAs. Ineffective erythropoiesis is another cause of anemia in which erythroblasts in bone marrow decay, such as anemia in β-thalassemia and myelodysplastic anemia. In this category, miRNAs in the early erythroid cells are released to circulating blood and serum level of miR-451a demonstrated disease severity [13]. Thus, miR-451a might be useful to determine treatment strategy. In malaria, severe anemia is complicated with acute hemolysis and ineffective erythropoiesis [44], although the exact mechanism is not yet clear. Although our clinical observation was still small size and more accumulation of data is recommended, our data showed that pre-clinical modification was not large to modify the hemolytic disease-specific elevation of miR-451a and miR-486.

At the analytical steps, there is more difficulty for the clinical application of miRNA biomarkers. RT-qPCR and next-generation sequencing (NGS) are approaches to detect miRNAs [45]. Those are expensive and relatively time-consuming. Several new technologies are making it possible to detect miRNAs: isothermal amplification, digital PCR, surface plasmon resonance methods [45]. The isothermal method is rapid and useful for the diagnosis of infectious diseases. Our group reported the method that utilizes a high-sensitivity enzymatic assay for detecting inorganic pyrophosphate to detect amplified PCR products [46], which is expected for use in biochemical automatic analyzer in the clinical laboratory in the hospital. Utilizing miRNAs for diagnosis of hemolytic diseases along with rapid and cheaper technologies are recommended in near future.

## 5. Conclusions

Our data showed that pre-analytical steps of blood sampling modify serum miRNA signatures mainly by destruction of RBCs. Rapid separation of serum from sampling tubes is recommended for reliable miRNA data. Serum miR-451a significantly increases in patients with hemolytic anemia more than the levels caused by pre-analytical modification. Thus, pre-analytical modification of erythroid cell-derived miRNAs (miR-451a and miR-486) does not disturb utilization of the two miRNAs as hemolytic disease-specific biomarkers.

## Figures and Tables

**Figure 1 jcm-10-05045-f001:**
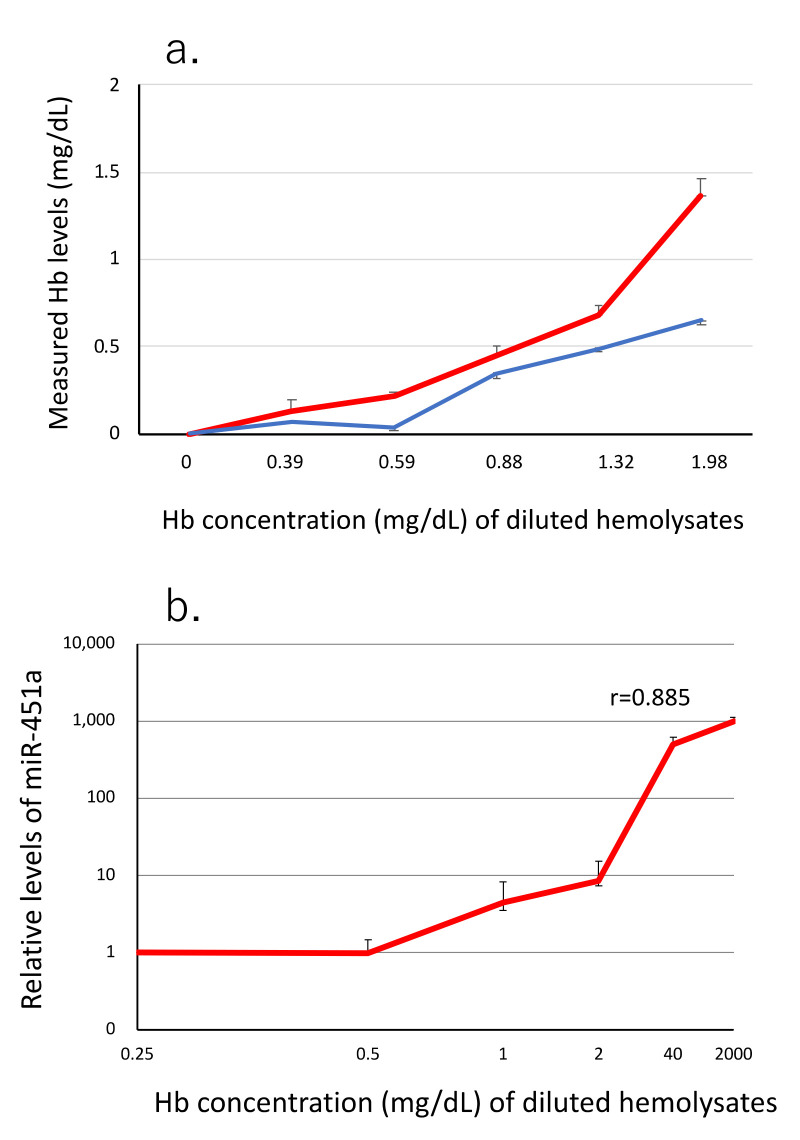
Sensitivity of miR-451a for hemolysis. (**a**) Comparison of sensitivity to detect Hb between the oxyhemoglobin method (red line with SEM bar, *n* = 3) and the optical densitometry method (blue line). Measurement was conducted using sequentially diluted hemolysates with the concentration of 0.39 to 20 mg/dL of hemoglobin. Asterix indicates statistically significant elevation from the blank level (*p* < 0.05). (**b**) miR-451a assay to detect hemolysis: Relative values of measured levels of miR-451a correlated well with Hb concentration (r = 0.885, *n* = 3, bar: SEM).

**Figure 2 jcm-10-05045-f002:**
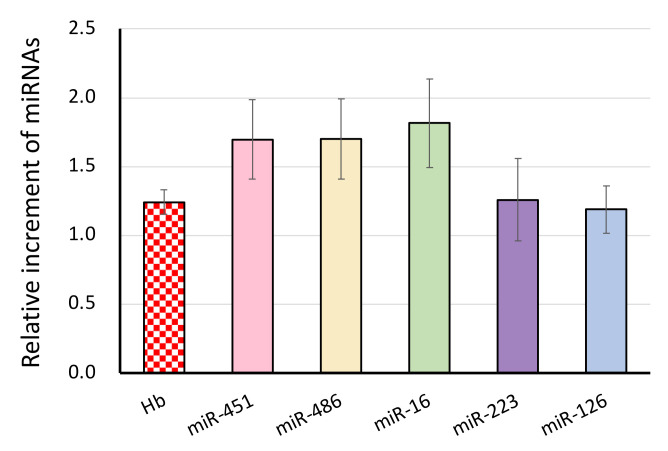
Relative increments of miRNAs after ex vivo blood drawing. The Hb level was measured using 414 nm OD. miRNAs were assayed with the RT-qPCR method. Each value (average ± SEM, *n* = 8) shows fold increments of miRNAs after blood drawing from the first blood samples.

**Figure 3 jcm-10-05045-f003:**
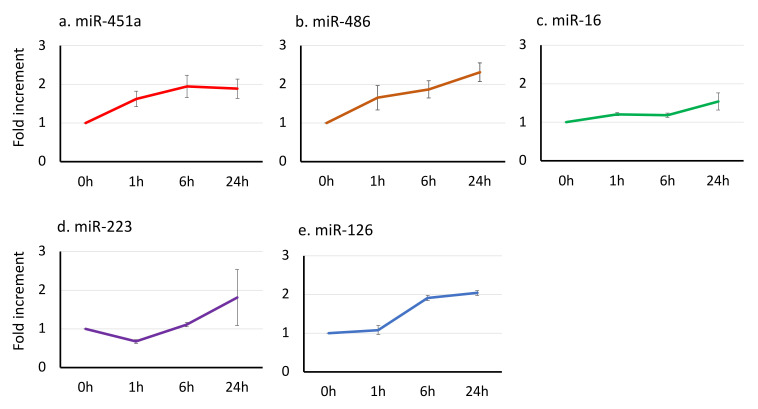
Relative increments of miRNAs before serum separation at 0, 1, 6, 24 h lag-time. miRNAs were assayed with the RT-qPCR method. Each value (average ± SEM, *n* = 8) shows fold increments of miRNAs after blood drawing at each time points, (**a**) miR-451a, (**b**) miR-486; (**c**) miR-16; (**d**) miR-223; (**e**) miR-126.

**Figure 4 jcm-10-05045-f004:**
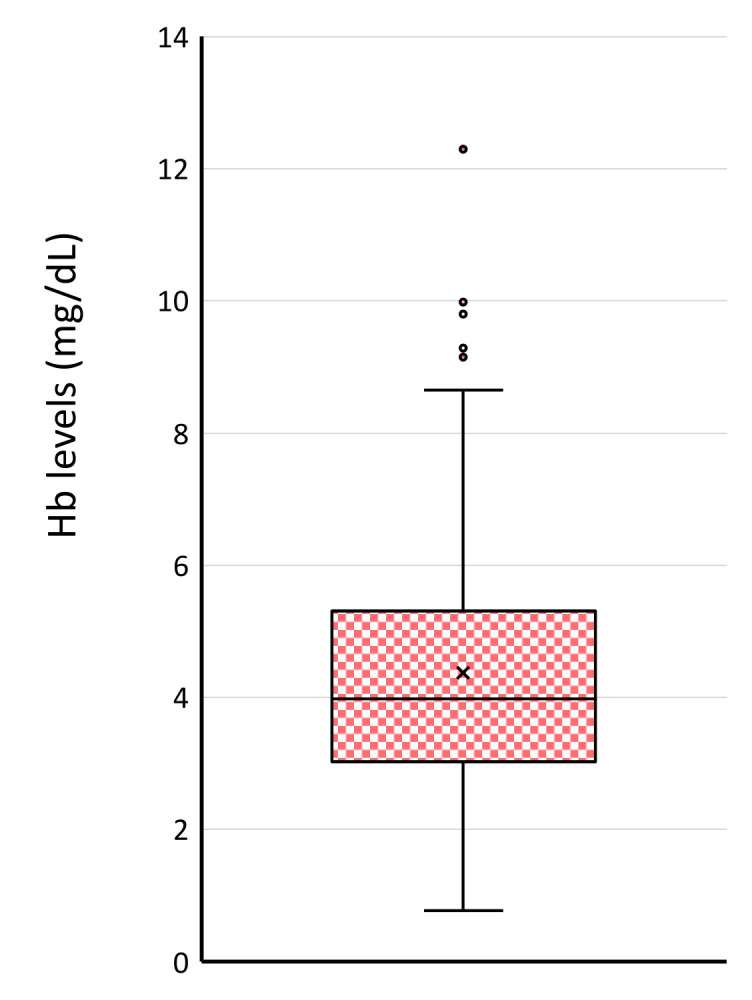
Free Hb levels in 116 leftover serum samples in clinical laboratory. Oxyhemoglobin was measured using 414 nm optical density.

**Figure 5 jcm-10-05045-f005:**
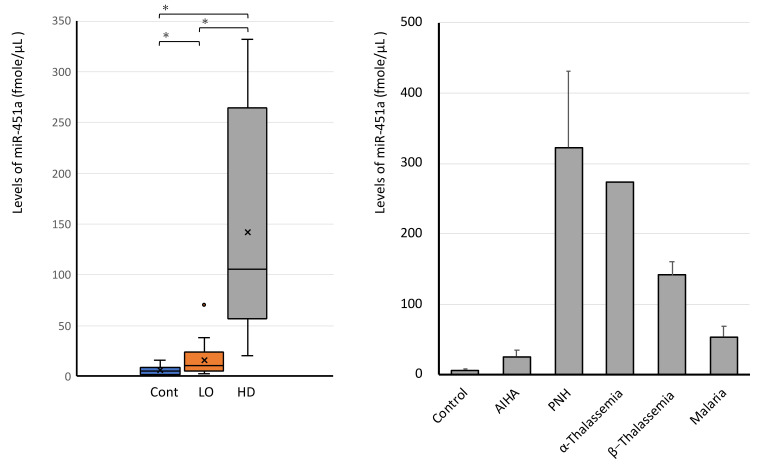
Serum miR451a in 20 patients with hemolytic diseases. The levels of serum miR-451 are shown as calculated values (fmol/µL, f: femto, 10^−15^). Cont: control subjects (*n* = 10), LO: leftover samples (*n* = 30), HD: hemolytic diseases (*n* = 20), AIHA: autoimmune hemolytic anemia, PNH: paroxysmal nocturnal hemoglobinuria. *: significant statistical differences (*t*-test, *p* < 0.05).

**Table 1 jcm-10-05045-t001:** Measurement of hemolysis indexes in 116 leftover serum samples.

Hemolysis Index	Average ± SEM (*n* = 116)	Range	Normal Range
Hb	4.37 ± 0.19	0.77–12.3	<5 (mg/dL)
LD	162.8 ± 3.80	75–325	120–220 (U/L)
AST	22.9 ± 1.52	8–160	10–35 (U/L)
K	4.2 ± 0.05	3.0–5.4	3.5–4.5 (mmol/L)

Hb: hemoglobin, LD: lactate dehaydrogenase, AST: aspartate aminotransferase, K: potasium.

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
