# Peer review of "Pre-Analytical Modification of Serum miRNAs: Diagnostic Reliability of Serum miRNAs in Hemolytic Diseases"

_jcm, 2021, doi:10.3390/jcm10215045_

Round 1
Reviewer 1 Report
Dr Takada and coullegues present in this manuscript a study investigating the value of miRNA as a biomarker for hemolysis. The investigators found that miR-451a was specifically associated with hemolysis and was not influenced by the time lag between sampling and processing.
While the study was well conducted and presented, there are a few issues that needs to be addressed.
The authors evaluated miRNA levels in patients with various hemolytic processes and compared the results to blood samples of controls with various time lags between sampling and processing. That could be a reasonable choice, but it is not clear how the authors regarded other potential pre-analytical errors apart from time for example pressure and duration of the tourniquet placing.
The study group comprised of several subgroups of patients- some with hemoglobinopathies, some with infections and others with immune mediated disease differing significantly with regards to the mechanism of hemolysis. . The authors needs to address this problem in the discussion and explain why they think that their results are relevant for any hemolytic disease,
Another important point is the clinical rather than the scientific value of this study. The authors should discuss what clinical value their study provides. As we see many patients with hemolysis, which patients would benefit from an expensive and mostly unavailable test
Author Response
Dear Reviewer 1,
We give many thanks for your worthy comments. We have done revisions according to your comments and suggestions. We sincerely hope that revised version is now ready for your consideration.
Sincerely Yours,
Tsukuru Umemura, M.D., Ph.D.

Reviewer 2 Report
The manuscript investigates several circulating miRNAs as potential blood disease biomarkers. The level of miRNAs is measured under several pre-analytical parameters that might affect the level of miRNAs. Overall, although certain conditions can change the level of some miRNAs as much as a factor of two due to pre-analytical hemolysis. Nevertheless, the level of miRNAs deriving from red blood cells (miR-451a and mi-R486) are much larger in disease groups compared to the normal groups. Thus, these miRNAs were proposed as potential biomarkers for hemolytic diseases, and some technical recommendation for the sample preparation was proposed to minimize the pre-analytical modification of the miRNA levels.
Some comments:
- The introduction part is too short and does not sufficiently introduce the background studies to convey the significance of the problem. The first part of the conclusion (lines 255-265) should better move to the introduction.
- Supp Figure 1 should be moved to the main manuscript, changed to Figure 1 and the other Figures moved accordingly.
- Supp Figure 1 - The scale on the X-axis was missing.
- Figure 1, the X- and Y- scales do not match. The Y-axis is obviously a log scale but the X-axis is neither linear nor logarithmic. The X-scale should be changed to the log scale.
- Line 146: MiRNAs rich in RBCs --> MiRNAs that are ubiquitous in RBCs
- Line 159: 1.65 by -> 1.65-folds by
- Figure 3 is too small to see clearly. The numbers on the axis scale should be enlarged and the axis should be labeled.
- Some explanation of the results should be added under topics 3.3 and 3.5 (or in the discussion section) rather than just showing the results. In this version, no explanation was provided in the discussion section either.
- Several typos or grammatical errors should be corrected such as method spectrophotometry method (line 102), was a sensitive enough (line 135), signatur (line 140), dell-derived (lines 146,279), optic density (line 177) etc.
Author Response
Dear Reviewer 2,
We give many thanks for your worthy comments. We have done revisions according to your comments and suggestions. We sincerely hope that revised version is now ready for your consideration.
Sincerely Yours,
Tsukuru Umemura, M.D., Ph.D.
